# Incidence of Emergence Delirium in the Pediatric PACU: Prospective Observational Trial

**DOI:** 10.3390/children9101591

**Published:** 2022-10-21

**Authors:** Eva Klabusayová, Tereza Musilová, Dominik Fabián, Tamara Skříšovská, Václav Vafek, Martina Kosinová, Michaela Ťoukálková, Adéla Vrtková, Jozef Klučka, Petr Štourač

**Affiliations:** 1Department of Paediatric Anaesthesia and Intensive Care Medicine, University Hospital Brno and Faculty of Medicine, Masaryk University, Kamenice 5, 62500 Brno, Czech Republic; 2Department of Simulation Medicine, Faculty of Medicine, Masaryk University, Kamenice 5, 62500 Brno, Czech Republic; 3Department of Applied Mathematics, Faculty of Electrical Engineering and Computer Science, VSB—Technical University of Ostrava, 70800 Ostrava, Czech Republic

**Keywords:** emergence delirium, emergence agitation, pediatric anesthesia, PAED score, Watcha score, RASS scale

## Abstract

Emergence delirium (ED) is a postoperative complication in pediatric anesthesia characterized by perception and psychomotor disorder and has a negative impact on morbidity in the form of maladaptive behavior, which can last weeks after anesthesia. Patients with developed ED present with psychomotor anxiety, agitation, and are at higher risk of unintentional extraction of an intravenous cannula, self-harm and nausea and vomiting. The described incidence of ED varies between 25–80%, with a higher prevalence among children younger than 6 years of age. We aimed to determine the incidence of ED in pediatric patients (>1 month) after general anesthesia in the post-anesthesia care unit (PACU), using Paediatric Anaesthesia Emergence Delirium (PAED) score, Watcha score and Richmond agitation and sedation scale (RASS). The incidence of ED was the highest in the PAED score with cutoff ≥10 points (89.0%, *n* = 1088). When using PAED score >12 points, ED incidence was 19.3% (*n* = 236). The lowest incidence was described by Watcha and RASS scores, 18.8% (*n* = 230) vs. 18.1% (*n* = 221), respectively. The threshold for PAED ≥10 points seems to give false-positive results, whereas the threshold >12 points is more accurate. RASS scale, although intended primarily for estimation of the depth of sedation, seems to have a good predictive value for ED.

## 1. Introduction

Emergence delirium (ED) is a complication occurring in the postoperative period in pediatric anesthesia that can develop frequently. There is no precisely determined definition of ED at present. It is described as a dissociative state of consciousness in which the child is irritable and crying, non-cooperative, incoherent and inconsolably moaning, or trashing about in bed [1]. ED usually develops in the first 15–30 min after lifting anesthesia. Its duration is variable but self-limiting [2]. ED might be associated with severe adverse events, which might include the accidental removal of intravenous catheters or drains, damage to incision sites, injury to the patient or personnel, all of which also lead to increased nursing requirements and increased parental anxiety and dissatisfaction with perioperative care [3]. The long-term effects of ED are not well described, but it is known that it might cause physiological implications such as separation, anxiety, sleep and eating disorders [4].

The described incidence of ED is unfortunately not clear and is burdened with a significant publication bias, as it depends on the definition of ED and the screening tool used. Various resources state that the incidence of ED is between 25% and 80%, with a higher prevalence among young children under 6 years of age [5,6]; however, ED can occur in even younger children and infants [7,8].

Although the pathophysiological basis for the development of ED is not entirely clear, a disturbance of the homeostatic processes in the brain is thought to be an important factor [9]. There are many risk factors which can precipitate ED; some of the most frequently described include a younger age, male gender, prolonged use of benzodiazepines or rapid emergence after general anesthesia in a non-familiar environment [9,10,11]. Both a longer duration of surgery and anesthesia seem to be negative contributing factors to ED [12]. The treatment consists of the administration of sedatives at reduced dosing, e.g., propofol, midazolam, ketamine or dexmedetomidine [13].

The only tool that is validated for ED diagnostics is the Paediatric Anaesthesia Emergence Delirium (PAED) score, which assesses five different characteristics: (1) the child makes eye contact with the caregiver, (2) the child’s actions are purposeful, (3) the child is aware of his/her surroundings, (4) the child is restless, and (5) the child is inconsolable [2]. However, there is no clear consensus on the use and evaluation of the PAED score. The authors consider the use of a threshold above ≥10 points to be specific to the diagnosis of ED. Previous studies have shown that a threshold of ≥10 points shows falsely high positive results; therefore, a threshold of >12 points might be more beneficial [6]. Moreover, the score is quite complicated and is considered lengthy by many anesthetists; it is also not entirely clear when after the end of general anesthesia the scoring should be initiated.

Although not validated for use of ED diagnosis in pediatric patients, there are other scoring systems that can be used, such as Cravero and Watcha scores. These scores are easier and faster to use and showed good sensitivity and specificity [14]. The Richmond agitation and sedation (RASS) score is a validated tool used for the evaluation of the depth of sedation and agitation in the intensive care unit (ICU). Apart from other scoring systems for the evaluation of sedation, this also includes the evaluation of the agitation component [15,16].

We conducted this observational trial to determine the incidence of ED in pediatric patients after general anesthesia the primary aim was to determine the incidence of emergence delirium in the post-anesthesia care unit (PACU) using Paediatric Anaesthesia Emergence Delirium (PAED) score, Watcha and Richmond agitation and sedation scale (RASS).

## 2. Materials and Methods

The trial was designed as a prospective observational trial conducted at the tertiary pediatric anesthesia centre—Department of Paediatric Anaesthesiology and Intensive Care Medicine, University Hospital Brno and Faculty of Medicine, Masaryk University, Czech Republic in the term from 1 September 2020 until 30 June 2021. The trial was initiated after the approval of the Ethics Committee of the University Hospital Brno, Jihlavská 20, 62500, Brno, Czech Republic (Approval Number: 09/09/2020; chairperson: Pharm. Dr. Kozáková; date of approval: 9 September 2020) and after registration on clinicaltrials.gov (NCT04531020). Written informed consent was not needed due to the observational design of the trial. The inclusion criteria were patients who had received general anesthesia and were admitted to the PACU within the defined time period. Patients aged from 0 to 1 months were excluded from the trial.

As a gold standard for the diagnosis of ED, we used the PAED score. Together with PAED, we also evaluated Watcha and RASS scores. To see whether there was a difference in incidence when different cutoff values of PAED score are used, we decided to set two thresholds as being positive for the ED: cutoff ≥10 points and cutoff >12 points. Our hypothesis was that a PAED >12 points is more accurate for the diagnosis of ED. We also used the Watcha score, due to its ease of use and a good sensitivity for ED diagnosis. The RASS scale was implemented for two reasons. Firstly, although there is not a clear consensus on when the scoring should be initiated, the use of RASS score could help to exclude the effect of the residual sedation after general anesthesia. Secondly, even though the RASS scale was never used to evaluate ED, we thought that it might be effective in ED diagnosis because it includes the agitation component. 

Paediatric Emergence Delirium (PAED) score, Watcha score and Richmond agitation and sedation scale (RASS) were measured at 0., 5., 10., 15., and 20. min after PACU admission. In case the RASS score upon admission was −3 or lower, the measurement began at the time at which the patient first made eye contact with the caregiver (obtaining RASS ≥ −2). Emergence delirium was defined as a PAED score greater than or equal to 10 points and/or a Watcha score over 2 and/or RASS over 1 minimally in one of the measurements.

Where ED developed, its duration was measured. The number of therapeutic interventions, as well as the dose of administered sedatives, was recorded. The incidence of postoperative nausea and vomiting (PONV) and the use of antiemetics, the type of anesthesia induction (inhalation vs. intravenous), the type of anesthesia (total intravenous ansthesia (TIVA), combined, inhalational), the length of the surgery and the type of the surgery were reported.

Numerical variables are expressed as the median and interquartile range; categorical variables are presented with the absolute frequencies and relative frequencies (%). The incidence of ED is estimated with the relative frequency and corresponding confidence interval. For between-group comparisons, we used the Chi-Square test of independence (or for equality of proportions), or the Mann–Whitney test. For an illustration of the progress of scoring systems in time, multiple boxplots combined with jitter plots were used. However, it was not possible to use the preferred graphical tool—paired boxplots—due to bad readability caused by the large sample size. The analysis was performed with maximum available data, the significance level was set to 0.05, and the analysis was performed using R software (version 4.2.1).

## 3. Results

Overall, 2241 patients underwent general anesthesia for surgical or diagnostic procedures, were hospitalized at the PACU and were assessed for eligibility. For organizational reasons, 890 patients were not included, and 1421 patients were assessed for enrollment. A total of 1222 patients were included in the final statistical analysis. Out of these patients, 199 patients were excluded, due to a scoring error (*n* = 178) and inconsistent data (*n* = 21). Figure 1 illustrates the flowchart of patient enrolment. The demographics of the cohort are described in Table 1.

Overall, 53.4% (*n* = 653) patients underwent inhalational anesthesia induction; intravenous anesthesia induction was used in 44.6% (*n* = 545) patients. The type of anesthesia used was balanced (84.9%, *n* = 1037), inhalational (8.9%, *n* = 109), combined (2.8%, *n* = 34) and total intravenous anesthesia (0.8%, *n* = 10). The most common type of surgery was general surgery, followed by otorhinolaryngology.

The median length of surgery/procedure was 50 min (IQR 35–65 min) and the median length of hospitalization at the PACU was 40 min (IQR 30–55 min). The incidence of PONV was 3.3% (*n* = 40). Antiemetics were administered to 4.6% (*n* = 56), including patients with a known personal history of PONV, for which the antiemetics were prophylactically administered.

The results of the primary outcome (the incidence of emergence delirium) are shown in Table 2. The incidence of ED varied significantly depending on the scoring system used. The highest incidence of ED occurred when a PAED score with cutoff ≥10 points was used, where 89.0% (*n* = 1088) of patients were described as delirious. Out of these patients, 78.9% (*n* = 232) met the criteria in at least two scoring systems and 54.8% (*n* = 161) met the criteria in all three scoring systems (PAED > 12, RASS or Watcha). Only 21.1% (*n* = 62) patients were diagnosed by only one of these criteria (PAED > 12: *n* = 26, RASS: *n* = 19, Watcha: *n* = 17). These results are graphically illustrated in Figure 2.

Regardless of the scoring system used, patients with observed ED were younger, and the incidence of ED was higher in patients who underwent inhalational anesthesia induction compared to intravenous induction. The highest incidence of ED according to the type of surgery has been diagnosed in stomatology procedures and plastic surgery (Table 3). The assessment of delirium over time during hospitalization in the PACU, according to each scoring system, is shown graphically in Figure 3. The estimated cumulative duration of ED in cases where it developed is shown in Table 4. In total, 20.7% (*n* = 253) of patients required pharmacological treatment of ED, out of which 77.5% (*n* = 196) were given only one dose of sedative and the remaining required the repeated administration of sedatives. The most prevalent sedative used for the treatment of ED was propofol (91.3%, *n* = 231) in median dose 0.83 mg/kg (min 0.16; max 3.12), followed by midazolam (5.6%, *n* = 14) in median dose 0.08 mg/kg (min 0.03; max 0.22).

## 4. Discussion

The incidence of ED in our patient population was 18.1–89.0% depending on the scoring system used, in accordance with previous studies [6].

In our trial, we wanted to demonstrate the discrepancies in ED incidence depending on the scoring system used and to evaluate which score is the most accurate, while excluding the effect of possible residual sedation. Although ED is most prevalent in pre-schoolers, it can also occur in young children and infants (e.g., Cohen et al. described ED incidence in children >2 months old) [8]. Although PEAD scale was originally vlidated for children aged from 18 months to 6 years, we are currently lacking a screening tool that could describe ED incidence throughout the whole age spectrum of pediatric patients. The PAED scale was repeatedly used in both children older than 6 years and children younger than 18 monhs [7]. For these reasons, we performed the study in a wide age spectrum of pediatric patients, because the fact that ED is the most common in pre-schoolers does not necesarrily mean that it cannot develop in different age categories.

Although the PAED scale is effective in precluding pain and its sensitivity is relatively high, the false-positive rate is also quite high [2]. Moreover, the evaluation of the scale is perceived by many anesthetists as complicated and lengthy. Another problem when evaluating ED by the PAED score arises from the use of different scales (a simplified version where only two or three characteristics are evaluated [17], as well as different cutoff values for ED diagnosis [18,19]. The maximum score is 20; according to the authors of the PAED scale, a cutoff of ≥10 points is positive for ED diagnosis [2]. However, when looking closely at the scale description, the child who is sleeping will score 12 points and therefore will falsely positively appear as ED-positive (the child makes eye contact with the caregiver—4 points, the child’s actions are purposeful—4 points, the child is aware of his/her surroundings—4 points, the child is restless—0 points, the child is inconsolable—0 points). In a few studies, the threshold was set even higher, to ≥16 points, but, in our opinion, this has no relevant justification [17,19]. The results of our study have shown a good correlation of the incidence of ED when using PAED >12 points, Watcha and RASS score, which could further justify the PAED cutoff >12 points.

The incidence of ED when using a PAED score with a cutoff ≥10 points vs. cutoff >12 showed a statistically significant difference. This corresponds with the false positive findings by Meyburg et al., where the optimal sensitivity and specificity were achieved with the cutoff >12 points, while the cutoff ≥10 points was considered to be set too low [6]. The incidence of ED by the Watcha, RASS scale and the PAED score with a cutoff >12 points were comparable; moreover, the cumulative duration of ED was the same in 86.3% (*n* = 139) of these patients, from which we can deduce a good ability of these scoring systems to assess ED over time.

The Watcha scale seems to be a very practical tool for ED evaluation; because it is only a four-point scale, it is very easy, staff-friendly and fast to use. Moreover, the overall high sensitivity and specificity compared to other scales are consistent with the results in our patient cohort [14]. 

The RASS score is not validated for the diagnosis of ED, but it is validated for use in evaluating sedation and agitation in critically ill children from age >1 month [15,16]. It is a simple and fast tool, with which many anesthetists and intensivists are familiar. Its advantage is that it can be used in both sedated patients (and ventilated, in case of intensive care unit (ICU)) and aware patients, and it can evaluate agitation as well. Our results showed that the incidence of ED defined by RASS >1 in at least one measurement was comparable with Watcha and PAED scoring systems. This suggests its possible use for diagnosing ED after general anesthesia; however, further studies would be necessary to validate the scale for this particular use.

The incidence of ED was higher in patients who underwent inhalational anesthesia induction compared to intravenous induction with propofol, similarly to previously described results [20]. It is believed that anesthetics with low blood/gas partition coefficient cause are a risk factor for ED development, sevoflurane the most [21]. Although it is believed that the rapid emergence can also negatively impact the incidence of ED, the more rapid extubation time with desflurane has not been associated with a higher incidence of ED when compared to sevoflurane [18]; however, the duration of ED is shorter after desflurane anesthesia [4].

Pain is also an important factor that can contribute to ED development and effective treatment of pain may reduce its incidence and severity [22,23,24]. However, Cravero et al. demonstrated that ED can develop in patients not undergoing painful procedures [25]. We consider one of the limitations of our study to be that we did not attempt to verify the level of pain at defined time intervals along with the delirium scoring, although the PAED scale reflects items that represent pain (2 items: the child is restless, the child is inconsolable). The heterogeneity of the patient population, the surgical and diagnostic procedures, as well as the unicentric and observational character of the trial, are the limits of our study.

Perioperatively used medications that may also negatively contribute to the development of ED include anticholinergic medications, benzodiazepines and inhalation anethetics with low solubility, especially sevoflurane [10,11,26]. The effect of midazolam on ED prevention is not clear, because its effect depends not only on the timing of administration before anesthesia, but also on the route of administration 19 October 2022 12:58:00. However, dexmedetomidine seems like a promising drug that can reduce the incidence of ED when administered preoperatively, as it showed superiority to midazolam and opioids [27].

In the case of the development of ED, the most-used drug for its treatment was propofol with a median dose of 0.83 mg/kg. Propofol appears to be effective for the treatment of ED, but the evidence for its beneficial effect is not entirely conclusive [11,19]. A recent meta-analysis from Wang et al. suggests that a combination of pharmacologic therapies might be of greater effectiveness than monotherapy (e.g., combination of dexmedetomidine, midazolam and antiemetics or a combination of propofol, midazolam and antiemetics) [13]. As monotherapy, dexmedetomidine, high-dose melatonin or nalbuphine showed good effects as well [13,19,28]. Dexmedetomidine in regular dosing does not cause respiratory centre depression, while having an effect as an anxiolytic, improving the child’s cognitive function and contributing to dose-dependent ED attenuation after general anesthesia [27]. Apart from pharmacologic treatment, several non-pharmacologic interventions to reduce the chance of ED development were described. The incidence of ED closely correlates with the perioperative anxiety of the child, so the techniques which might help to reduce anxiety are of great benefit, e.g., preoperative visiting of the operation room, video distraction during mask induction of anesthesia, tablet-based interactive distraction or recording mother’s voice to help arouse the patient [21,29].

This trial in a statistically large cohort of patients demonstrated heterogeneity in the incidence of ED depending on the scoring system used. In our opinion, the PAED score, although the only valid tool to evaluate pediatric emergence delirium, has several drawbacks, is complicated, and gives false-positive results at the recommended threshold of ≥10 points, which is consistent with results from other studies. In contrast, the Watcha score compared to the PAED >12 points showed a good correlation, and is more user-friendly and faster to evaluate. RASS scale, to which anesthetists are often accustomed from assessing sedation, might also be a reliable tool to detect ED, but further research is needed to validate this.

## 5. Conclusions

In this observational trial, the incidence of pediatric emergence delirium was heterogenous, ranging from 18.1% to 89.0% depending on the diagnostic tool used for ED. The cutoff of PAED >12 points correlated with the Watcha and the RASS scale.

## Figures and Tables

**Figure 1 children-09-01591-f001:**
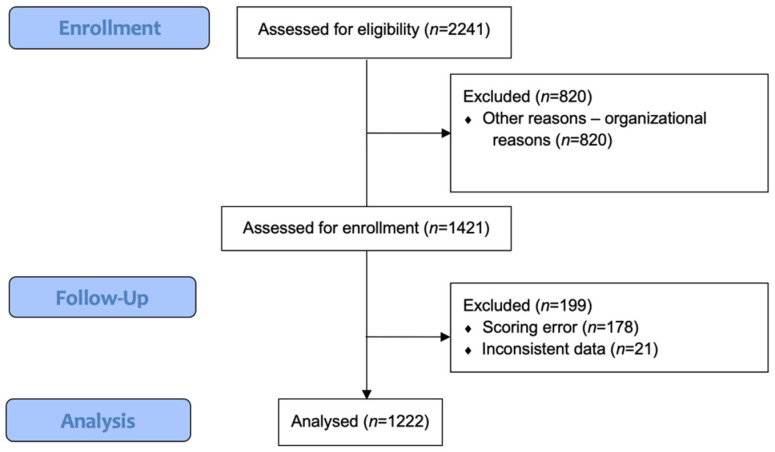
Flowchart.

**Figure 2 children-09-01591-f002:**
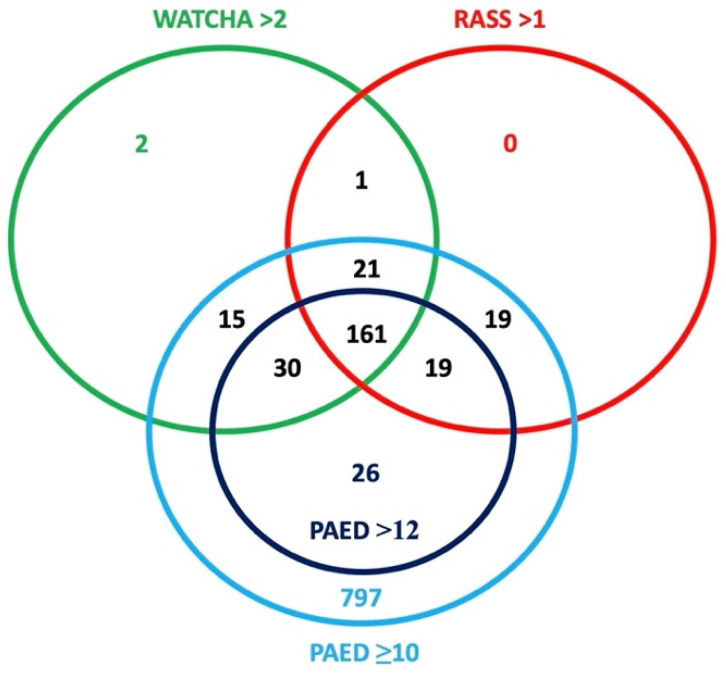
Graphical illustration of the incidence of ED using different scoring systems.

**Figure 3 children-09-01591-f003:**
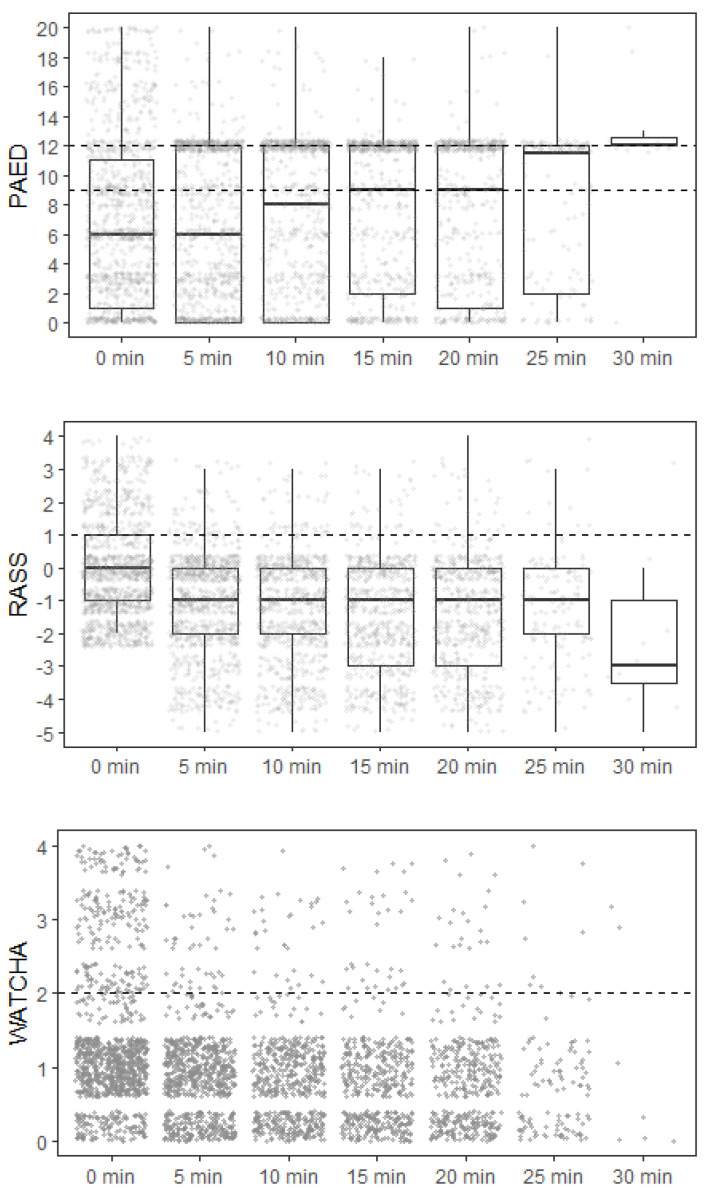
The assessment of delirium over time during hospitalization in the PACU. The dashed line for each scoring system indicates the threshold above which the patient was identified as positive.

**Table 1 children-09-01591-t001:** Demographics.

	Median (IQR) or *n* (%)
Age (years)	8 (4; 13)
Weight (kg)	30.0 (18.0; 53.0)
Type of surgery	
General surgery	370 (30.2)
Otorhinolaryngology	333 (27.3)
Ophthalmology	219 (17.9)
Orthopaedics	182 (14.9)
Stomatology	58 (4.8)
Plastic surgery	30 (2.5)
Diagnostic procedures	20 (1.6)
Neurosurgery	4 (0.3)
Combined types of surgery	2 (0.2)
Unknown	4 (0.3)
IQR—interquartile range	

**Table 2 children-09-01591-t002:** Emergence Delirium (in at least 1 measurement).

	Delirium Criteria (In at Least 1 Measurement)			
	PAED ≥ 10	PAED > 12	RASS > 1	WATCHA > 2
Number of patients	1088	236	221	230
Rel. frequency (95%CI)	89.0% (87.1–90.8)	19.3% (17.1–21.7)	18.1% (15.9–20.4)	18.8% (16.6–21.4)
		** *p* ^a^ **		
PAED ≥ 10 vs. RASS > 1 vs. WATCHA > 2		<0.001		
PAED > 12 vs. RASS > 1 vs. WATCHA > 2		0.736		
PAED ≥ 10 vs. PAED > 12		<0.001		

**^a^ *p***-value of the Chi-square test for equality of proportions, PAED—Paediatric Emergence Delirium score; RASS—Richmond agitation and sedation scale, CI—confidence interval.

**Table 3 children-09-01591-t003:** Incidence of ED according to the PAED, Watcha and RASS score compared with demographic variables.

Delirium Criteria (in at Least 1 Measurement)	PAED > 10			PAED > 12			RASS > 1			WATCHA > 2		
	Yes	No	** *p ^a^* **	Yes	No	** *p ^a^* **	Yes	No	** *p ^a^* **	Yes	No	** *p ^a^* **
Median of age in years (IQR)	7 (4; 12)	13 (9; 15)	<0.001	3 (2; 6)	10 (5; 13)	<0.001	3 (2; 5)	9 (5; 13)	<0.001	3 (2; 6)	10 (5; 13)	<0.001
Type of surgery			<0.001			0.001			<0.001			<0.001
General surgery	327 (88.4)	43 (11.6)	-	65 (17.6)	305 (82.3)	-	52 (14.1)	318 (85.9)	-	58 (15.7)	312 (84.3)	-
Otorhinolaryngology	289 (86.8)	44 (13.2)	-	69 (20.7)	264 (79.3)	-	74 (22.2)	259 (77.8)	-	73 (21.9)	260 (78.1)	-
Ophthalmology	209 (95.4)	10 (4.6)	-	49 (22.4)	170 (77.6)	-	45 (20.5)	174 (79.5)	-	44 (20.1)	175 (79.9)	-
Orthopaedics	137 (75.3)	45 (24.7)	-	19 (10.4)	163 (89.6)	-	13 (7.1)	169 (92.9)	-	18 (9.9)	164 (90.1)	-
Stomatology	57 (98.3)	1 (1.7)	-	21 (35.0)	39 (65.0)	-	20 (34.5)	38 (65.5)	-	20 (34.5)	38 (65.5)	-
Plastic surgery	24 (80.0)	6 (20.0)	-	8 (26.7)	22 (73.3)	-	10 (33.3)	20 (66.7)	-	11 (36.7)	19 (63.3)	-
Diagnostic procedures	19 (95.0)	1 (5.0)	-	3 (15.0)	17 (85.0)	-	5 (25.0)	15 (75.0)	-	4 (20.0)	16 (80.0)	-
Type of anesthesia induction			<0.001			<0.001			<0.001			<0.001
Inhalation	608 (93.1)	45 (6.9)	-	202 (30.9)	451 (69.1)	-	191 (29.2)	462 (70.8)	-	194 (29.7)	459 (70.3)	-
Intravenous	442 (81.1)	103 (18.9)	-	29 (5.3)	516 (94.7)	-	23 (4.2)	522 (95.8)	-	30 (5.5)	515 (94.5)	-
Length of surgery (min)	45 (35; 65)	55 (41; 75)	<0.001	50 (40; 65)	45 (35; 65)	0.482	50 (40; 65)	45 (35; 65)	0.185	50 (40; 65)	45 (35; 65)	0.054

The values represent the absolute frequency and relative frequency (%). **^a^ *p***-values were obtained with the Mann–Whitney test or the Chi-Square test of independence.

**Table 4 children-09-01591-t004:** The estimated cumulative duration of ED.

Estimated Cumulative Delirium Duration	Delirium Criteria (in at Least 1 Measurement)			
	PAED ≥ 10	PAED > 12	RASS > 1	WATCHA > 2
≤5 min	715 (65.7)	184 (77.4)	166 (75.1)	174 (75.7)
(5; 20] min	140 (12.9)	52 (22.6)	51 (23.1)	54 (23.5)
>20 min	233 (21.4)	-	4 (1.8)	2 (0.8)

The numbers represent the absolute frequency and the relative frequency (%).

## Data Availability

The data presented in this study are available on request from the corresponding author.

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
