# Peer review of "Incidence of Emergence Delirium in the Pediatric PACU: Prospective Observational Trial"

_children, 2022, doi:10.3390/children9101591_

Round 1

Reviewer 1 Report

The manuscript “Incidence of emergence delirium in the PACU: prospective observational trial” is an interesting study. However, there are some points that need to be elucidated.

Major revision

1.       The type of population that needs to supplement this study in the title.

2.       The purpose of this study was to assess the incidence of postoperative ED in children, were the subjects included consecutively?

3.       Eight hundred and twenty patients who met the inclusion criteria were excluded from the cohort population of this study, and it is very likely that the occurrence of ED events in the missing cases would change the results of this study. Sensitivity analysis is recommended for missing patients.

4.       For patients with incorrect scores or incomplete data, it is difficult to determine the incidence of ED events, and sensitivity analysis is required to verify the stability of the conclusions.

5.       The age span of the children in this study is very large, and stratified analysis should be carried out for different age groups to explore whether the conclusions are stable in different ages.

Reviewer 2 Report

Please see attached file with detailed comments and suggestions.

Round 2

Reviewer 1 Report

I have no suggestions for the authors.

Author Response

Dear reviewer, 

thank you for your review and your decission.
